# Understanding Spontaneous Symbolism in Psychotherapy Using Embodied Thought

**DOI:** 10.3390/bs14040319

**Published:** 2024-04-12

**Authors:** Erik Goodwyn

**Affiliations:** 1Department of Psychiatry and Behavioral Sciences, University of Louisville, Louisville, KY 40202, USA; 2College of Medicine, University of Kentucky, Lexington, KY 40506, USA; erik.goodwyn@uky.edu; 3The Billings Clinic, Billings, MT 59105, USA

**Keywords:** embodied metaphor, depth psychology, cognitive and affective neuroscience, psychobiology, hypnotherapy, evolutionary biology, code biology

## Abstract

Spontaneous, unwilled subjective imagery and symbols (including dreams) often emerge in psychotherapy that can appear baffling and confound interpretation. Early psychoanalytic theories seemed to diverge as often as they agreed on the meaning of such content. Nevertheless, after reviewing key findings in the empirical science of spontaneous thought as well as insights gleaned from neuroscience and especially embodied cognition, it is now possible to construct a more coherent theory of interpretation that is clinically useful. Given that thought is so thoroughly embodied, it is possible to demonstrate that universalities in human physiology yield universalities in thought. Such universalities can then be demonstrated to form a kind of biologically directed universal “code” for understanding spontaneous symbolic expressions that emerge in psychotherapy. An example is given that illustrates how this can be applied to clinical encounters.

## 1. The Embodied Nature of the Mind

Whereas early psychoanalysts such as Freud and his followers often saw mind and body as fairly interconnected, early experimental and clinical psychology, seeking a more rigorous science of the psyche, broke away from psychoanalytic traditions and maintained a fairly rigid boundary between them, as was seen, for example, in the prominence of behaviorism. Drawing inspiration from advancements in computer technology, however, the extreme agnosticism of the mind found in behaviorism eventually gave way to the cognitive revolution of the 1960s and 1970s, which saw an explosion of models of how the mind worked. Nevertheless, cognitive psychology continued the trend of treating the mind as quite separate from the body/brain. Here, the mind was viewed as a manipulator of arbitrary, abstract symbols. Such symbols were no more than *representations* of the body and environment, meaningless in themselves, to be manipulated via various mechanisms to achieve a desired environmental outcome [1]. Meanwhile, the things the mind was representing (body and environment) were seen as utterly devoid of mental properties in themselves.

This rigid dichotomy between mind and body, however, began to erode in the 1980s with the emergence of various sorts of *embodied* thought. Theorists and clinicians began to recognize that the old view, which saw an isolated mind operating on an essentially mindless body, was incomplete and inaccurate. Affective neuroscience [2,3,4] slowly built a vast empirical basis for an embodied view of the emotions as strongly innate cross-species and evolution-derived organizers of cognition and behavior. Affective neuroscience also saw consciousness itself as existing in a multilayered way at different levels. Evolutionary psychology [5], for example, began to clash with the Standard Social Science Model and its disembodied characterization of the mind. Cognitive neuroscience endorses an evolutionary and embodied view with its intense focus on the brain following the results of various neuroimaging techniques [4]. Still other trends in psychology have emerged, such as that of Nummenmaa et al. [6], who were able to link basic emotions with culturally universal sensory body maps, further linking cognition, body, and emotion. Mounting evidence of so-called ‘psi’ phenomena [7,8] which outright *contradict* any dualism between mind and body, has continued to accumulate.

On the clinical side, newer kinds of psychotherapy have come to reject the dualistic conceptualization as well. Psychodynamic therapies that essentially link mental states and ideas to the body to treat chronic pain and other somatoform disorders [9] have continued to gain empirical support. Somatic therapies [10,11] theorize that the body has its own kind of intentionality, perception, and mental capacity independent of egoic awareness. Psychopharmacology continues to advance, piercing the mind-body “barrier” within psychiatry. Neuropsychoanalysis [12], based on Neo-Freudian psychoanalysis merged with neuroscience, fundamentally rejects a dualistic view of mind and body. Psycho-systems analysis [13,14] also firmly rejects a dualistic view of mind and body, while advocating for a strongly monistic perspective that incorporates psychobiology, genetics, hypnotherapy, and analytical psychology.

Philosophy itself began to see a rising interest in panpsychism and other non-dualistic ontologies, attacking the disembodied view of the dualistic paradigm directly as well as its cousin, the physicalist paradigm [15]. Finally, perhaps the most devastating blow to the above disembodied view of mental contents came from the growing literature on embodied cognition, starting with Lakoff and Johnson’s famous work on cognitive metaphor theory [16]. More on this later.

Clinical psychotherapy material can, of course, vary widely, as each person has their own individual history and life course. That said, we are all members of the species *Homo sapiens*, a species with a set of universally shared biological characteristics. As will become evident below, an embodied view of cognition means that the universalities of the body can translate to certain universalities of cognition. This fact becomes more evident in the study of *spontaneous* thought–i.e., thoughts, images, and even narratives that emerge into our minds unbidden by wilful intent.

## 2. Spontaneous Thought, including Dream Content

It has long been held in various psychoanalytic schools that spontaneously emergent thoughts and images are not random but can reveal depths of intentionality and meaning from unconscious dimensions of the mind. This conclusion finds some empirical support in the literature on spontaneous thought [17]. Also referred to as “mind wandering” or “task-unrelated thought”, the literature on spontaneous thought incorporates neuroscience, phenomenology, and other disciplines. Akin to the early psychoanalysts, these researchers find that such mental contents do, in fact, appear to be non-random, purposeful, organized, and often highly functional. Moreover, it appears to have

“considerable benefit for our day-to-day functioning and general contentment–affording sense-making and the ordering of recent events, anticipations of and projections into the future, and a starting point for some of our creative ideas.”([18], p. 487; see also [19]).

A principle finding from this literature is that the brain, rather than being a reactive organ that only passively responds to external stimuli, has a multilayered, intrinsically active overall structure that is intensely dynamic and creative even in the absence of input. As mentioned, spontaneous thoughts are not merely chimerical wish-fulfilments. Rather, they are focused on emotionally salient personal concerns and involve memory consolidation and future planning, most of which are directed at sorting out current concerns [20]. Though spontaneous thought can be associated with dysfunctional processes such as rumination, it typically subserves many *functional* processes such as facilitating semantic knowledge consolidation, emotional processing, planning, and pattern recognition [21].

Research on this subject has expanded tremendously in the past decade. In fact, some researchers have labeled the modern era of neuroscience the “era of the wandering mind” [22]. Roughly one-third to one half of thought is spontaneous [19], and this doesn’t include the particular kind of spontaneous thought represented by dream content. Unsurprisingly, dreaming has been classified as a *type* of spontaneous thought [17] that shares much overlap with waking spontaneous thought. Like waking spontaneous thought, dreams are biased toward affectively salient personal concerns, and the process comes ‘on line’ by around age 7–10 [23]. The overlap of waking and dreaming spontaneous thought can also be observed neuroscientifically, as the same regions involved in spontaneous thought are recruited (albeit more intensely) during REM dreaming [24,25].

Analyzing dreams is traditional in psychoanalytic traditions; however, there has been disagreement about whether or not such content is largely occupied with *disguising* deeper thoughts [26] or largely *revelatory* of deeper processes. Carl Jung, for example, disagreed with Freud and proposed the latter, describing dream contents as a “*spontaneous self-portrayal, in symbolic form, of the actual situation in the unconscious*” ([27], para 505, emphasis original). Similarly, Bion proposed that dreams transform unprocessed primary experiences into workable thoughts, memories, and mental growth [28]. Both of these early psychoanalysts proposed that this process occurs in both waking and sleeping states. By contrast, Freud [26] originally argued that dreams *disguised* inner realities rather than revealing them.

In a recent comprehensive review by Roesler [29], however, a large body of empirical literature on dreaming was reviewed, and it was found that Freud’s older idea that dreams disguise meaning has not aged very well. Neo-Freudian thought has therefore since adapted its overall position [12]. In any case, the balance of empirical literature on dreams shows that they have a strong tendency for meaning *creation*, which can be observed in the manifest dream content, provided one recognizes the symbolic nature of such content. More on symbolism later.

## 3. Spontaneous Thought Is Organized by Affect and Is Embodied

Affective states influence the amount and direction of spontaneous thought content [30], and spontaneous thoughts are affectively centered and sensitive to current goals [17]. Zedelius and Schooler [31] review studies that show that intentionally attending to spontaneous thought (a common psychodynamic technique) even has a mood-regulating function. This research showed that it is the *image-heavy* content of spontaneous thought, rather than verbal description, that is most effective at mood regulation. They found that only when subjects imagined first-person, image-heavy scenarios did their mood improve, regardless of whether the scenario was positive or negative ([31], p. 241). More on these insights later.

Much of the above characteristics and functions of waking spontaneous thought have been found for dream-state spontaneous thought as well. That is to say, dreaming has been identified as serving several functions, including mood regulation, problem solving, adaptation, furthering mastery, fostering insight, and anxiety reduction [29]. Dream researchers Kramer and Glucksman [32] also showed how multiple dreams in a single person can be observed to be continually working and reworking major emotional themes, even to the point that independent evaluators could identify immediate and long-term emotional issues just from dream content. Notably, Roesler [29] states in his review:

“This reworking of memory contents in the dream is therefore a highly structured, rule-governed and goal-directed reworking process that operates largely unconsciously, extensively coordinates various domains of mental functioning, but can also only take place while there is no new mental input of the kind that occurs in the waking state.”([29], p. 314)

Spontaneous thoughts, then, appear to arise all the time and seem to be focused on taking life experiences and re-organizing them into either affectively directed autobiographical narratives (i.e., “memory”) or into symbolic expressions of one’s current or upcoming life situation (dreams and fantasies). Importantly, vast literature in affective neuroscience finds that affective states themselves correlate with the activity of highly conserved (i.e., invariant) subcortical brain structures that are *universal* in humans and have strong analogues in other animals [2,3,4]. Spontaneous thoughts in general, then, are strongly influenced by deeply embodied and universal processes. However, what are they about? What is their purpose? The answer appears to be that they are continuously working on context construction, and their specific manifestations have a particular character related to our shared universal physiology.

## 4. Spontaneous Thoughts Seek Meaning

A growing number of dream researchers have proposed that dreaming processes experiences by attempting to connect them with other experiences in memory, effectively meaning-making via context construction (reviewed in [29], pp. 303–308). Dreaming functions have been proposed to serve the development, preservation, and reintegration of psychic organization [33]. Neuroscientifically, spontaneous thought (including dreaming) is characterized as related to the so-called “default mode” network, a brain network consisting of midline, lateral parietal, and mediolateral temporal brain regions. The default mode network is associated with internally directed attention, representations of self, and autobiographical memory, among other functions [34]. This network operates “in the background” but is particularly prominent when one is doing nothing in particular. It is exploratory in nature and heavily weighted toward pattern recognition and creativity, along with many independent measures.

Stan and Christoff [18] argue that the numerous meaning-making benefits of spontaneous thought might justify changing the term “mind-wandering” to “mind-ordering”, given its functions. Moreover, growing evidence points to spontaneous thoughts being critical in the construction of *personal identity and meaning*, suggesting a role in “reflecting on the broader meaning and implications of personal experiences, thereby contributing to the construction, maintenance, and update of an individual’s life story.” ([30], p. 187). These findings independently corroborate the speculations of earlier psychologists and psychoanalysts that proposed an *identity-consolidating function* of dreaming [35,36,37]. Kliner et al. [38] in their review of the literature on spontaneous thought conclude that such thoughts tend to be involved with goal tracking, planning, creative problem solving, reviewing past experience, memory consolidation, and aligning oneself with goal attainment, though when in excess, spontaneous thoughts can be ruminative, engage in excessive daydreaming, or even be involved in dissociation.

Nevertheless, the implication from the literature of spontaneous thought (especially of the dream-derived variety) is that such thoughts are very often *emotionally focused symbolic expressions of one’s current life situation and/or future plans*, much like the early psychoanalysts speculated. Only now is such speculation supported by independently verified empirical research. However, are such thoughts truly symbolic? How can we test this? To really explore this question, we must take a closer look at the literature on embodied cognition and metaphor theory, which examine symbolic thought more generally. As will be seen, this literature helps to better understand the commonly metaphorical *nature* of spontaneous thought, which may then aid clinicians in the *interpretation* of such clinical material.

## 5. An Intrinsic Integrating Process

Early psychoanalysts such as Kohut, Jung, and Bion (among others) hypothesized processes strongly akin to what the foregoing research suggests, i.e., that the mind intrinsically tries to incorporate and integrate events of the past into a coherent narrative, and moreover, that this intrinsic process appears to aim toward emotional regulation, updating of self-identity, future planning, and contextualizing of trauma. The important result is that *spontaneous products of the imagination* seem to be a key player in this process. Its meaning-making functions, as manifested in the spontaneously emergent elements, appear to seek toward these goals via the construction of emotionally focused narratives that seem to aim for an as-if depiction of the current life situation as a whole.

In summary, the mind contains a continuously operating, ongoing process that is not directed by everyday conscious will or direction, and this process produces spontaneous expressions of the imagination. Such products are affectively directed and, as we will see, often metaphorical and embodied in nature.

The problem for clinicians, therefore, is to determine how we may properly *interpret* such expressions. If they are metaphors, what are they metaphors *of*? Since they are unconsciously created, this question can be challenging, as we will not be privy to the metaphor-creating process. This is where the study of embodied cognition is extremely useful.

## 6. Embodied Cognition

The fundamentally metaphorical nature of thought in general, and spontaneous thought in particular, is a central finding of embodied cognition research. Rather than view all cognition as disembodied sign manipulations emerging solely in response to verbal instruction or cultural observation, cognitive linguists challenged the older cognitive paradigm using a number of crucial observations of real-world, everyday language [1,16,39]. The literature of embodied cognition and cognitive linguistics, in fact, shows that much of our thought can be described not in meaningless signs but in terms of *embodied metaphors*, which are themselves defined as:

“frame-to-frame mappings across conceptual domains…linguistic metaphors are surface reflections of those conceptual mappings…from correlations between co-occurring embodied experiences; for example, Happy Is Up, Sad Is Down; More Is Up, Less Is Down; Affection Is Warmth.”([1], p. 776).

For example, basic-level categories and spatial relations across many languages require universal primitives that reference the human body [40]. These metaphors map abstract or difficult to describe/comprehend domains onto embodied visuospatial domains that are easy to comprehend, such as the LOVE IS A JOURNEY mapping ([16], capitalization by convention). Complex conceptual metaphors are composed of *primary* metaphors, which do not further decompose. Most relevant for our purposes, a subset of these primary metaphors are deeply *embodied*, arising from physical correlates of human emotion, and are hence ubiquitous cross-culturally, such as when anger is described as a hot liquid (“boiling over with anger”, “letting off steam”, etc.). In other words, some primary metaphors are reliably and cross-culturally emergent due to universalities in human physiology. More examples of these kinds of primary metaphors are

UNDERSTANDING IS SEEING: “I can’t understand his point of view”, “that argument is murky”, “his thought is very clear”.

HAPPY IS UP: “flying high with good feelings”, “on cloud nine”

SAD IS DOWN: “feeling down in the dumps”, “I’m *depressed* today”

CONCEPTUAL HARMONY IS PHYSICAL BALANCE: “These ideas don’t work because they are unbalanced”

ATTACHMENT LOSS IS COLD: “he gave me the cold shoulder”, “she left me out in the cold”

In contrast to the classic disembodied-meaningless-symbol theory, modern cognitive linguists argue that “conceptual knowledge is embodied, that is, it is mapped within our sensory-motor system [which] characterises the semantic content of concepts in terms of the way that we function with our bodies in the world…Abstract reasoning in general exploits the sensory-motor system.” ([41], pp. 456–473)

There is significant empirical support for embodied metaphor theory, coming from action-sentence compatibility studies, eye-tracking studies, hand-prime studies, gesture-in-learning studies, and neuroimaging studies on sensorimotor activation during metaphor processing and mental imagery processing [39]. As mentioned, this literature supports the idea that even abstract, complex metaphors are *compositions* of simpler metaphors. These simpler metaphors are mappings of ideas onto embodied visuo-spatial and kinesthetic body movements. Take the ANGER IS HEAT metaphor, for example. The source of this concept is the human body itself and its physiological responses to the emotion of anger ([39], p. 8). Other primary metaphors use kinesthetic sense (UNDERSTANDING IS GRASPING, “I can’t grasp this subject”, or CONTROLLING IS GRASPING “you need to get a grip on this situation”), which is naturally embodied. Such a metaphor would make no sense to a whale or a rhinoceros, but it makes perfect sense for a handy primate such as *Homo sapiens*.

## 7. The Symbolic Dimension of Embodied Cognition

An important feature of conceptual embodied metaphor theory is that all metaphors have an *ineffable core of meaning* that cannot be expressed except via other metaphors [16,42,43]. This finding is precisely *why* metaphors are useful, in that they are able to depict and signify that which is difficult to verbalize. This applies not just to fancy expressions of deep mystery (such as poetry or interpretive dance) but also to more prosaic ones such as DANGER IS DARKNESS, as is seen in the evocative imagery of “the dark forest” found in so many fairy tales. “Danger”, after all, cannot ever literally be “darkness”, and yet the metaphor is powerful and viscerally resonant for poor humans with lousy night vision.

One consequence of this feature of metaphors is that *spontaneous* metaphors will have this quality to them as well, i.e., it will be difficult to verbalize their meaning without the use of other metaphors. This feature, however, is not due to any necessarily mysterious, unknowable characteristic of unconscious cognition but rather to the *metaphorical* manner in which it is often expressed. Metaphors are an effective and compact way to consolidate information into an easily digestible form to guide cognition and behavior.

## 8. The Universal “Code”

The findings of embodied cognition reveal that many spontaneous thoughts are complex metaphors that express our current emotionally-focused situation or future plans. Such spontaneous content emerges merely by nature of the brain’s primary function to organize the vast amount of information pouring into it every day into a compact, guiding narrative of some kind. The use of metaphor, in turn, is merely an effective tool to do this job more efficiently.

Complex metaphors, according to this theory, will be composed of primary metaphors working in concert. I contend, furthermore, that a significant subset of the primary metaphors will be essentially *innate* in origin. By “innate” I simply mean that they are self-organizing within the mind of anyone possessed of a human body. They do not depend on cultural learning or observation to acquire–they simply occur to us naturally by virtue of being human beings in a generically terrestrial environment. What learning is involved in them is merely self-teaching. For the rest of this essay, I will propose which primary metaphors I think are universally emergent and how this helps with the interpretation of clinical material. Understanding this will help us develop a universal “code” for understanding spontaneous symbols in clinical settings.

I define such universally emergent primary metaphors as ***innate mappings***. Innate mappings do not need to be observed or acquired via didactic learning. They are derived solely from our embodiedness as human beings in the world. Some specific examples of innate mappings include (borrowing and elaborating from [16]:

KNOWLEDGE/SAFETY/HAPPINESS IS LIGHT (which also implies UNMANIFEST/DANGER/SADNESS IS DARKNESS)

POWER/HAPPINESS IS UP (which implies WEAKNESS/SADNESS IS DOWN)

CONCEPTUAL HARMONY IS SYMMETRY/BALANCE

AFFECTION/ENJOYMENT/LIFE IS WARMTH

ANGER/PASSION/LUST IS HEAT/HOT LIQUID

COMPLEX PROCESS IS A CONSCIOUS BEING (i.e., personification)

EMOTIONS ARE FACIAL EXPRESSIONS

UNDERSTANDING IS GRASPING

WILLING IS MANIPULATING WITH THE HANDS

EMOTIONAL SEPARATION IS COLD

I provide more proposed innate mappings below. For now, it should be noted that this theory could be criticized on the grounds that such mappings could simply be learned in the conventional sense (i.e., via cultural instruction or observation) and so could potentially not be universal. Indeed, being taught such mappings provides sufficient conditions for their acquisition in the developing mind. Nevertheless, it remains to be seen whether or not such learning constitutes the *necessary* conditions for their emergence. It could still be possible that we would acquire these mappings anyway. Demonstrating that culture/observation are not necessary for the emergence of these mappings, however, is quite difficult since it would ideally involve studying humans who grew up devoid of human contact. Such experimental conditions would naturally be lethal since we are so thoroughly social by nature. Nevertheless, there are clues that these mappings are not derived from cultural imitation but instead self-organizing in all human beings regardless of culture, and that attributing them entirely to learning is simply a kind of “nurture bias”.

For example, the above visuospatial metaphors arise even in the congenitally blind [42], and they can be found in spontaneous gestures and signs in signed languages as well [44]. Nevertheless, one might argue they were acquired via verbal instruction from sighted people. To rule that out, persons born blind were presented with novel visuospatial metaphors (i.e., “he ascribes to a photoshopped version of reality”). They still had no trouble understanding them, despite having *no* shared experiential referents. This and other data led researchers to conclude: “to a great extent, conceptual metaphors are universal knowledge structures that are associated to the structure of our body and of our exchanges with the world. As such, they impose strong constraints on the space of possible innovations in metaphoric production.” ([44], p. 7). Such results demonstrate that observation and instruction *do not* seem necessary to acquire the visuospatial metaphors I label as innate mappings.

## 9. A Specific Innate Mapping Example

Let’s consider ANGER IS HEAT, a mapping I claim is innate. In English, there are many common sayings that utilize this primary mapping, such as “hot under the collar”, “a real hothead”, “boiling over with anger”, etc. I argue that this mapping is innate because, when we feel anger, our temperature increases along with rising blood pressure. Skin flushing from anger gives the sensation of hotness by virtue of our human physiology. We do not need verbal instruction or observation to make this association; we are biologically predisposed not only to make it ourselves but to understand it on a visceral level. If the human body did not have the above physiological responses, this mapping would not make sense or naturally arise in *Homo sapiens*.

If, however, universal physiology/predispositions are *not* necessary to form this mapping, then it is likely just idiosyncratic to English, and it should not be found with great regularity in other languages. Without any biological constraints or organization, one could form any sort of arbitrary mapping imaginable for anger without dissonance. Thus, cross-cultural study of conceptual metaphors can falsify the hypothesis that this mapping is innate. If all that is required is verbal instruction or cultural observation to produce ANGER IS HEAT, then the mapping should not realistically be found cross-culturally, and its opposite might even be found.

When examining unrelated languages, however, this mapping does occur in English, Chinese, Japanese, Hungarian, Tahitian, Chickasaw, and Wolof ([45], pp. 156–163). Cognitive linguist Zoltán Kövecses reviews many other cross-cultural examples such as this and concludes that “My view is that, given the universal real physiology, members of different cultures cannot conceptualize their emotions in a way that contradicts universal physiology…[though] they can choose to conceptualize their emotions in many different ways within the constraints imposed on them by universal physiology.” ([45], p. 165, emphasis original). Elsewhere, he states that “Feeling states have an irreducible and probably universal psychobiological basis that accounts for many similarities in the conceptualization of emotions.” ([45], p. 187).

None of this discounts the importance of culture, which can elaborate some innate mappings over others. For example, Zulu utilizes ANGER IS HUNGER (both universal situations) to organize many metaphorical sayings, whereas this mapping is not commonly found in English. That said, if someone said “consuming the whole village with his anger”, an English speaker is still very likely to understand it because both anger and hunger are universal. This does mean that for any given language, some mappings may only exist in potential. That said, no culture will use the *opposite* of the innate mappings–i.e., HAPPINESS IS DOWN, PRIDE IS SHRUNKEN POSTURE, CONCEPTUAL HARMONY IS ASYMMETRY, CONTROLLING IS RELEASING ONE’S GRIP, etc.

Furthermore, many primary mappings *are* entirely cultural in origin and not innate at all. Chinese uses HAPPINESS IS FLOWERS IN THE HEART, and Zulu uses ANGER IS GRINDING CORN ([45], pp. 167–169) to organize many sayings in those languages. These are non-innate, unique cultural primary mappings, since flowers and corn do not derive from the human body or its species-specific responses to generic environmental features (the criteria I use for innateness). Rather, both of these are derived from unique cultural traditions. In the construction of spontaneous thought, of course, both innate and non-innate mappings will be used together to create all sorts of affective symbols.

Below is a list of several more mappings that I propose are innate, based on universal body experiences, developmental milestones, and/or physiological responses. These do not require cultural observation or didactic learning and are self-organizing. These arise even in individuals born blind, deaf, or otherwise impaired [46]:


Temperature


AFFECTION/ENJOYMENT/LIFE IS WARMTH

ANGER/PASSION/LUST IS HEAT/HOT LIQUID

SEPARATION IS COLD


Light/Vision


KNOWLEDGE/SAFETY/HAPPINESS IS LIGHT (which also implies UNKNOWN/UNMANIFEST/DANGER/SADNESS IS DARK)


Body Orientation


CONCEPTUAL HARMONY IS SYMMETRY/BALANCE

CONCEPTUAL INTEGRATION IS A JOINING OF MALE AND FEMALE

AFFECTION IS CLOSENESS

CONCEPTUAL SIMILARITY IS CLOSENESS


Personification


COMPLEX PROCESS IS A CONSCIOUS BEING (i.e., personification)

EMOTIONS ARE FACIAL EXPRESSIONS

ORIGINATING PRINCIPLE IS A PARENT/ANCESTOR


Hands


UNDERSTANDING IS GRASPING

WILLING IS MANIPULATING WITH THE HANDS

EMOTIONAL DETACHMENT IS RELEASING GRASP


Body Integrity


EMOTIONAL TRAUMAS ARE PHYSICAL INJURIES

EMOTIONS ARE LIQUIDS

MENTAL CHAOS IS DISMEMBERMENT

VITALITY IS BREATH/WIND (“in-spir-ation”, etc.)

EMOTIONAL TURMOIL IS TUMBLING

VITALITY/VIVACIOUSNESS IS WATER (and opposite DEATH/DISENGAGEMENT IS DESSICATION)

FUNDAMENTAL PRINCIPLES ARE DOWNWARD/FEET

DEVELOPED PRINCIPLES ARE UPWARD/HEAD


Visuospatial


POWER/HAPPINESS IS UP (which implies WEAKNESS/SADNESS IS DOWN)

THE “MOST IMPORTANT” IS THE CENTER

HARMONIOUS INTEGRATION IS PHYSICAL SYMMETRY

BEAUTY IS PHYSICAL SYMMETRY

SENSORY INTENSITY IS VISUOSPATIAL VIVIDNESS (IN TERMS OF COLOR)

IMPORTANT IS BIG/UNIMPORTANT IS SMALL

EMOTIONAL EVENTS ARE PHYSICAL TRAJECTORIES (i.e., LOVE IS A JOURNEY)

CONCEPTUAL PROFUNDITY/ESSENCE IS DEPTH

CONCEPTUAL/SOCIAL TRANSITIONS ARE BOUNDARY CROSSINGS

## 10. Understanding Spontaneous Symbols in Psychotherapy

The foregoing implies, then, that the human body and its psychological embodiedness come with a large array of species-typical *responses* to recurrent and genetically anticipated species-specific situations (light, dark, basic emotions, up, down, hot, cold, basic social relations, etc.) that will be used in the construction of spontaneous thought to include many self-organizing innate mappings. Because they are derived from our human embodiedness in the world, they are ultimately derived from the human genome. Thus, given our natural tendency to form metaphorical mappings from an early age, innate mappings are a biologically transmitted “alphabet”, which the unconscious mind can use to compose all kinds of combined images using those innate mappings. Such compositions have the ability to arise in practically anyone, regardless of culture of origin or language, because they derive from our universal embodiness in the world. We do not need to acquire such symbols from cultural material or verbal instruction to have the capacity, indeed the *tendency*, for such symbols to occur to us spontaneously. This recognition provides clinicians with a set of tools to help make sense of spontaneous symbolic content, an example of which I will give below.

Note that this formulation also generates a set of falsifiable assertions: every one of the above symbolic associations that I theorize are innate can be subjected to a cross-linguistic analysis such as the one provided by Kövecsecs to determine if it is truly innate or not. Moreover, learning this code also helps us to interpret the images themselves, since knowledge of the origin of innate mappings means we will have a kind of “Rosetta stone” with which to analyze clinical material of all kinds. Note that, despite the fact that complex metaphors might be composed of many innate mappings, each such spontaneous product should be considered a whole greater than its parts, unique to the individual’s current situation, rather than “nothing but X”. It is always possible to create a new *combination* of innate mappings.

This formulation also explains why universal body-derived images can appear so mysterious and full of meaning despite their ineffable quality: as symbols, they are complex metaphors embodying powerful, but difficult-to-verbalize human feelings, perceptions, and expressions that are ultimately derived from our universal human embodiedness. Finally, this formulation explains how some mythic stories can appear so similar across vast cultural divides–*yet they are not identical*. Early psychoanalysts such as Carl Jung, for example, puzzled over this in many of his works, proposing “archetypes” as explanatory elements of the mind, though he struggled to define archetypes clearly. The above proposal might be considered a reimagining of archetype theory, only with much greater clarity of definition and supported by falsifiable empirical findings from spontaneous thought, embodied cognition, affective and cognitive neuroscience, evolutionary theory, and many other fields unavailable to the early psychoanalysts [46,47,48,49,50,51].

## 11. Clinical Example: How to Use the Code

Mr. A, a male atheist patient in his thirties with depression and anxiety symptoms, dreamt the night before beginning psychotherapy:

“I am in a tall, vertical apartment on the top floor, where I am to celebrate the birthday of “the great grandmother”, a very large and jovial woman. Then I find myself downstairs in bed with a small person with both male and female features, whom I am supposed to “date”. They send me alone into a desert in the night where I wander for long in the darkness, past dry, crumbled buildings, until I come upon a great dune. Crossing the dune, day comes, and I see hundreds of people playing on an ocean of extremely vivid, almost glowing blue. It is unfathomably deep and rich. I don’t merely guess, but KNOW that this ocean is “God”, and it is alive. She is our great mother. I wake up in tears because it is so beautiful.”

As mentioned in the foregoing research on spontaneous thought, we have the dreamer meeting personifications of different aspects of his lived experience, common in dreams [52]. A “date” with a character seems to imply an impending “blending” or “bringing together” of a pluripotent figure with the dreamer’s conscious egoic self, at the behest of a large (and hence important) ancestral “great grandmother”. So far, we can see and use the following innate mappings:

The great grandmother character combines:

COMPLEX PROCESS IS A CONSCIOUS BEING

IMPORTANT IS BIG (great grandmother)

ORIGINATING PRINCIPLE IS A PARENT/ANCESTOR (great grandmother)

And the trip downstairs to meet the dreamer’s date combines:

FUNDAMENTAL PRINCIPLES ARE DOWNWARD/FEET (going to the bottom floor)

COMPLEX PROCESS IS A CONSCIOUS BEING

CONCEPTUAL INTEGRATION IS A JOINING OF MALE AND FEMALE (used in two ways: in the character themselves and the implied joining of the dreamer and this character)

CONCEPTUAL SIMILARITY IS CLOSENESS—here viewed more dynamically as “bringing together”

Next, we see the dreamer engage in an emotional series of events that goes from darkness to light (i.e., “enlightenment”) and from dry, barren, abandoned, and desolate to a lush, deep ocean, overabundant with life. Here, many playful characters convey positive social engagement with the ocean, which the dreamer perceives as a divine, tangible, vivid, unifying god of limitless depth. This acquainting (or perhaps *reacquainting*) of the dreamer with this divine power brings about tears of joyful meaning.

Thus we see:

EMOTIONAL EVENTS ARE PHYSICAL TRAJECTORIES

KNOWLEDGE/SAFETY/HAPPINESS IS LIGHT

UNKNOWN/UNMANIFEST/DANGER/SADNESS IS DARK

DEATH/DISENGAGEMENT IS DESSICATION

Used in the journey:

CONCEPTUAL/SOCIAL TRANSITIONS ARE BOUNDARY CROSSINGS

The ocean itself is a memorable “character” that combines:

COMPLEX PROCESS IS A CONSCIOUS BEING (“God”)

SENSORY INTENSITY IS VIVIDNESS (IN TERMS OF COLOR)

EMOTIONS ARE LIQUIDS

AFFECTION/ENJOYMENT/LIFE IS WARMTH

VITALITY/VIVACIOUSNESS IS WATER

CONCEPTUAL PROFUNDITY IS DEPTH

Subsequent clinical material bore out an intense craving for meaning that struggled against a reductive and rigid thought pattern that relaxed considerably with therapy. Moreover, since it is loaded with innate mappings, this dream could have been dreamt by *anyone* belonging to the species *Homo sapiens* since the symbolism is derived from universal physiological invariants, *given the right inner and outer environment*. The culture-specific details such as apartments, clothing on the characters, and ruined buildings do not seem important to the overall symbolic expression, as they could easily be changed to other structures, clothing, or ruins, and the overall feeling and narrative would be identical.

## 12. Clinical Relevance

What is the significance of this dreamer experiencing a universal dream rather than a more idiosyncratic and personal dream prior to therapy? Only the recognition that not every clinical utterance is rooted in one’s personal development, but that some psychic contents are organized in a way that reflects our *species* history rather than our individual history, arising like a genetically guided story containing a kind of universal wisdom. Such contents should not be glossed over in therapy since they represent deeply *human* elements of one’s experience and the “genomic response” to their situation. This means that some psychic contents are not simply reflections of the personal past but are natural *species* (rather than individual) responses and attempts at adaptation to changing circumstances.

This kind of *psychological* expression and response is very similar to certain kinds of unlearned natural body responses. For example, when diving into water, humans and other air-breathing vertebrates exhibit a complex set of coordinated and innately guided responses that include peripheral vasoconstriction, slowed heart rate, diversion of blood to vital organs, and release of red blood cells from the spleen [53], all of which conserve oxygen. This response is not learned and has nothing to do with individual history. You could live your entire life and never know you had this ability until you dived into a body of water. Responsivity is built-in and ultimately derives from our *species* history as air-breathing vertebrates with numerous innate internal homeostatic mechanisms. It is not acquired through one’s individual childhood.

Hence, we should not rule out the likelihood that our history as extremely social primates has woven similar responses into our genomes when facing frequently encountered *psycho-social* challenges such as loneliness and isolation. For *Homo sapiens*, poor social adaptation would be lethal in the ancestral environment. Therefore, strong selection pressures existed for the majority of our hominin existence that could plausibly have generated a tendency to produce such responses in the form of spontaneous thoughts. Spontaneous thoughts tend to aim toward meaning, memory, and identity consolidation and represent one way in which the array of human innate responses could address such psychosocial problems through their effects on emotional regulation, social identity, and future planning. Innate mappings, moreover, are ideal to generate powerful, viscerally moving narratives that reflect commonly encountered survival (and hence, in our case, *socially oriented*) situations. Such narratives would come with a built-in understandability that crosses all cultural barriers.

From the foregoing, we can hypothesize that when certain important conditions are obtained (regardless of prior exposure to them), the genome may actively provide an epigenetic response by using the normally continuously active identity, memory, planning, and meaning consolidation processes already at work. The mechanisms of psychosocial situations triggering cascading shifts of genetic expression in the body and brain have been reviewed in great detail elsewhere [14,54,55,56,57,58]. Thus, it is likely that, via these sorts of epigenetic processes latent genetic responses may be “turned on” to produce such universally shaped narratives.

My hypothesis is that the more emotionally evocative the situation, the more likely a primordial/universal response will arise that uses the ‘inherited alphabet’ of innate mappings to orient spontaneous symbolic thought. Here, Mr. A.’s psyche took his learned experiences of apartments, oceans, and desert ruins and reorganized them into a powerful spiritual narrative of unity, community, and interconnectedness. Such a narrative likely depicted a deep need the dreamer had and provided a genetically anticipated adaptive solution to it. Similar to the diving response, they are likely attempts at adaptation, but they focus on our *social and intra-psychic adaptation* rather than our underwater swimming adaptation.

When we recognize that not everything that emerges spontaneously within the psyche is a reflection of early individual experiences, we realize that it is possible for the psyche to respond in a way that is based on ancestral rather than only local/individual determinants. This simple distinction necessitates the existence of *both* culturally learned/observed components as well as embodied, genomic, or “archetypal” sources of spontaneous thought construction. With the above model and “code”, we have a way to discern and more accurately interpret ancestral- vs. individual-derived material and recognize that, as many psychoanalytic schools teach, our unconscious symbolic-making capacities are often a valuable resource to access in therapy.

## Data Availability

Data are contained within the article.

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
