# Peer review of "Understanding Spontaneous Symbolism in Psychotherapy Using Embodied Thought"

_behavsci, 2024, doi:10.3390/bs14040319_

Round 1

Reviewer 1 Report

Comments and Suggestions for Authors

In my opinion, this is a very interesting article that familiarizes the reader with the possible role that the body plays in the process of psychotherapy.

The author presents evidence that spontaneous thought is organized by Affect and is Emodied. Personally, from clinical practice, I agree with the point of view: Emodied Cognition. I just wonder if there are codes of understanding specific to given nations, not for cultural reasons, but for historical traumas (which manifest themselves in changes in DNA in the form of epigenetic changes) that may modify the understanding of spotaneus symbols proposed by the author.

Author Response

I absolutely agree that this is a possibility, since we know that traumas can alter DNA methylation in germ cells and therefore be transmitted across generations. As a clinician, I imagine you have seen this, as have I--now we have a strongly plausible mechanism here. It would be a great subject to explore in a future article, I think. Thank you for this excellent suggestion!

Reviewer 2 Report

Comments and Suggestions for Authors

L 1-75: This is a solid introduction. Adding in less mainstream supports (The Jungian view of psyche, Gendlin's work on felt sense, Bateson's mind/nature, even going back to Bergson) would add more material, but your point is well made without it. 

Fix block quote L 84-90

L 76-126: Good general survey on dreams. Ellis has much regarding dreams and trauma (with references to other somatic theorists, like Bosnak's Embodied imagination). Gendlin also works with dreams similarly. As above, this is not at all necessary to add in or research here, but may provide fruitful avenues for later research or expansion for your general interest. 

L. 162: Missing "A"

162-196: This is fascinating stuff: you do a great job of weaving in relevant research and concepts and construct a great pathway for your reader. I'm riveted.

226: Block quote formatting edit needed.

196-268: Fantastic overview of Lakoff/Johnson/metaphoric systems.

300: Deeply appreciate the turn to innate core metaphors--in my mind, it'd be interesting to see where you'd take this alongside Jung's notion of the relation of instinct to archetype--not for this paper, not the same, but not totally unrelated, perhaps. 

346: I appreciate the detailed example of why Anger/Heat is innate.

346-449: Offering the list again is overall welcome, especially without a word limit. Breaking it into the conceptual categories adds a quick, useful reference guide for readers who return to this work. 

471-475: Gendlin's work here again would seem useful for further development--not needed in this paper, but for future projects.

476-488: I'm glad you brought Jung in relevantly here. In a future project, taking a step back to more clearly connect what you're offering to his sense of instincts (which builds on Freud's essay on vicissitudes, I suspect) that are experienced as meaning from the inside, but look "typical" on the outside, might be of interest.

489: What you do with this code, in the case study, is VERY impressive. 

562-564: This, again, is Jung's point throughout his works.

579-586: I appreciate this reworking of the larger point to advance your thesis. 

598: I love the notion of an inherited alphabet.

This is an incredible, innovative, well-researched and insightful development in psychology. I'm really grateful I had the chance to review and be exposed to it. Thank you. 

Author Response

Many thanks for these suggested references. I had to actively restrain myself from including more due to space constraints (especially with respect to Jung), but even still I am grateful for the suggestions of Gendlin and Bateson. 

Block quote fixed by adding double quotes at the beginning.

"A" added to L162.

I'm glad it's riveting!

Block quote fixed on L226

I'm a big fan of Lakoff's work on metaphors. 

L346--there are indeed many of these I outline in my other works. 

I have only a passing familiarity with Gendlin's work--a greatly appreciate this suggestion!

Much of this work began with an exploration into Jung's and Freud's ideas, definitely. I believe these two, especially Jung, were many decades before their time. As you mention in your comment regarding L562-4, I think Jung was actually spot on, only he really didn't have the needed empirical research or even terminology to really express it perfectly. Not to mention the zeitgheist of his day was that of the "blank slate" and generally very reductive views of the mind. 

I couldn't be more pleased at such a positive and thoughtful review, and again am deeply indebted to these suggestions for further reading.